# IgA Antiphospholipid Antibodies in Antiphospholipid Syndrome and Systemic Lupus Erythematosus

**DOI:** 10.3390/ijms23169432

**Published:** 2022-08-21

**Authors:** Tatiana Reshetnyak, Fariza Cheldieva, Maria Cherkasova, Alexander Lila, Evgeny Nasonov

**Affiliations:** V.A. Nasonova Research Institute of Rheumatology, Kashirskoye Shosse 34A, 115522 Moscow, Russia

**Keywords:** antiphospholipid antibodies, antiphospholipid syndrome, systemic lupus erythematosus, IgA antibodies to cardiolipin, IgA antibodies to beta-2 glycoprotein 1

## Abstract

**Objective:** To define the role of IgA antibodies to cardiolipin (aCL) and IgA antibodies to beta-2 glycoprotein 1 (anti-β_2_-GP1) in the development of vascular complications in patients with antiphospholipid syndrome (APS) and systemic lupus erythematosus (SLE). **Material and methods:** A total of 187 patients with one of the following diagnoses: primary APS (PAPS), probable APS, SLE with APS, and SLE without APS. The comparison group consisted of 49 patients with other rheumatic diseases (RD), the control group included 100 relatively healthy individuals (without RD, oncological pathology, and infectious diseases). All patients underwent standard clinical, laboratory, and instrumental examinations before being included in the study and during follow-up. The aPL study included the determination of IgG/IgM aCL, IgG/IgM anti-β_2_-GP1 by enzyme-linked immunosorbent assay (ELISA), IgG/IgM/IgA aCL, IgG/IgM/IgA anti-β_2_-GP1 by chemiluminescence analysis (CLA), and lupus anticoagulant (LA). **Results:** IgA aCL were detected in 75 (40%) of the 187 patients with APS and SLE, in none of the comparison group, and in 2 (2%) of the control one. IgA anti-β_2_-GP1 were detected in 63 (34%) of the 187 patients with APS and SLE, in none of the patients in the comparison group, and in one (1%) of the control group. The prevalence of IgA aCL and IgA anti-β_2_-GP1 and their levels were statistically significantly higher in patients with APS (PAPS and SLE + APS) than the levels in patients with SLE and those of the comparison and control groups (*p* < 0.05). IgA aCL and IgA anti-β_2_-GP1 were significantly associated with thrombosis in APS (χ^2^ = 4.96; *p* = 0.02 and χ^2^ = 4.37; *p* = 0.04, respectively). The risk of thrombosis was 2.04 times higher in patients with positive IgA aCL than in patients without these antibodies, as well as in patients with positive IgA anti-β_2_-GP1; it was twice as high as in patients without antibodies. There was a high specificity of IgA aCL and IgA anti-β_2_-GP1 for both the diagnosis of APS and its clinical manifestations, despite a low sensitivity. **Conclusions:** The study revealed a relationship of thrombosis and APS with IgA aCL and IgA anti-β_2_-GP1. There was a high specificity of IgA aCL and IgA anti-β_2_-GP1 (95% and 93%, respectively) for the diagnosis of APS with a low sensitivity (54% and 44%, respectively). There were no patients with isolated positivity of IgA aCL and IgA anti-β_2_-GP1.

## 1. Introduction

Antiphospholipid syndrome (APS) is an autoimmune multisystemic disorder characterized by recurrent thrombosis and pregnancy morbidity in patients with antiphospholipid antibodies (aPL) [1,2]. The serological markers of APS include IgG and/or IgM antibodies to cardiolipin (aCL) in serum or plasma, which are present in medium or high levels (>40 GPL or MPL units), IgG and/or IgM antibodies to beta-2 glycoprotein 1 (anti-β_2_-GP1), and lupus anticoagulant (LA), which are detected two or more times at a study time interval of at least 12 weeks [3]. The latter are included in the 1997 update of the American College of Rheumatology (ACR-97) criteria for systemic lupus erythematosus (SLE) [4]. Despite limited evidence, IgA aCL and IgA anti-β_2_-GP1 were included in the 2012 Systemic Lupus International Collaborating Clinics criteria (SLICC) [5]. The APS criteria were, in fact, aimed at harmonizing research, and they are widely used to establish a diagnosis and determine treatment options. This is important for patients with thrombosis and pregnancy morbidity, as this may require long-term, possibly lifelong anticoagulant therapy. Despite the existing criteria, the diagnosis and treatment of thrombotic APS are difficult tasks [6]. Although any of the above antibodies is sufficient to establish a diagnosis, the latter can be confirmed only after a stipulated period of 12 weeks. The diversity of pathogenic antibodies requires an expanded array of tests to significantly confirm or rule out APS in a patient with thrombosis. Limited resources associated with testing all patients for all three types of antibodies and retesting them to confirm the diagnosis remain an important factor in the management of patients until the final diagnosis and continuation of anticoagulant therapy.

Currently, IgA aPL are not part of the routine diagnostic testing for APS, and the role of IgA as a diagnostic marker is still a matter of debate [7]. The 13th International Congress on aPL recommended to study IgA anti-β_2_-GP1 and IgA aCL only in patients with clinical APS signs, negative for IgG/IgM anti-β_2_-GP1 and IgG/IgM aCL [8]. The issue of including IgA anti-β_2_-GP1 and IgA aCL in the classification criteria for APS continues to be discussed.

Objective: To determine the significance of IgA aCL and anti-β_2_-GP1 in the development of vascular complications in patients with APS and SLE. 

## 2. Results

IgA aCL were detected in 75 (40%) of the 187 patients. The frequency of IgA aCL was significantly higher in patients with APS: in 26 (49%) of 53 patients with primary APS (PAPS), in 5 (42%) of 12 with probable APS, in 35 (59%) of 59 with SLE + APS, in 9 (14%) of 63 with SLE, in none of the comparison group, and in 2 (2%) of the control one. 

As indicated in Figure 1, the median IgA aCL levels in patients with PAPS, probable APS, SLE + APS, and SLE were higher than those in the control group (*p* < 0.000001), higher in patients with PAPS, probable APS, SLE + APS than those in patients of the control group (*p* < 0.0001, 0.01; < 0.0001, respectively), and higher in patients with PAPS and SLE + APS than those in patients with SLE (*p* = 0.0001 and *p* < 0.0001, respectively). 

According to IgA aCL positivity, all patients were divided into two groups: (1) 75 patients with positive values (>18.9 CU) and (2) 112 with negative values (<18.9 CU). 

Verification of definite APS was statistically significantly associated with the positivity for IgA aCL (χ^2^ = 23.96; *p* < 0.0001) (Table 1). APS was detected in 61 (81%) patients with IgA aCL versus 51 (45%) patients without these antibodies (χ^2^ = 23.96; *p* < 0.0001; odds ratio (OR), 5.26; 95% confidence interval (CI), 2.63–11.11). The risk of developing APS was 5.26 times higher than that in patients without IgA aCL.

There was a significant relationship between IgA aCL and thrombosis (χ^2^ = 4.96; *p* = 0.02) (Table 1). Thrombosis was recorded in 53 (71%) of 75 patients with IgA aCL (χ^2^ = 4.96; *p* = 0.02). The risk of thrombosis in the patients with positive IgA aCL was 2.04 times higher than in those without these antibodies.

The diagnostic efficiency of IgA aCL was evaluated according to the ROC curves (Figure 2) depending on the presence of thrombosis (a), pregnancy morbidity (b), and a confirmed diagnosis of APS (c). The area under the receiver operating characteristic curve (ROC curve) was 0.791 [0.725–0.837] in thrombosis, 0.806 [0.725–0.886] in obstetric pathology, and 0.851 [0.803–0.889] in definite APS. All the results according to the ROC curves were significant (*p* = 0.0001). 

The sensitivity of IgA aCL for the diagnosis of APS, thrombosis, and obstetric pathology was 54, 39, and 32%, respectively. The specificity of IgA aCL was higher for these conditions (95, 88, and 93%, respectively).

The positivity of IgA aCL was significantly associated with the presence of IgG/IgM aCL; it was combined with the presence of IgG + IgM aCL in 34 (45%) of 75 cases, only with IgG aCL-in 71 (95%) and with IgM aCL-in 37 (49%). IgA aCL positivity was associated with the presence of anti-β_2_-GP1 in 100% of cases.

Fifty-four (29%) of the 187 patients were negative for IgG/IgM aCL and IgG/IgM anti-β_2_-GP1. Four (7%) of the 54 patients were found to be positive for LA. There was no isolated IgA aCL positivity in patients who were negative for classic aPLs.

IgA anti-β_2_-GP1 were detected in 63 (34%) of the 187 patients: in 22 (41%) of 53 patients with PAPS, in 4 (33%) of 12 with probable APS, in 28 (47%) of 59 with SLE + APS, in 9 (14%) of 63 with SLE in none of the patients in the comparison group, and in one (1%) of the control group. In patients with PAPS and SLE + APS, IgA anti-β_2_-GP1 was significantly more common than that in the control and comparison groups (*p* < 0.05). The incidence of IgA anti-β_2_-GP1 in patients with SLE + APS was higher than that in patients with SLE without APS (*p* = 0.02).

The median levels of IgA anti-β_2_-GP1 in patients with PAPS, probable APS, and SLE + APS were higher than those in the control and comparison groups (*p* < 0.0001) (Figure 3). The IgA anti-β_2_-GP1 levels were significantly higher in patients with PAPS and SLE + APS than in those with SLE (*p* = 0.0005 and *p* = 0.0000025, respectively), whereas the IgA anti-β_2_-GP1 values were higher in patients with SLE than in the control group (*p* = 0.009).

The patients were divided into two groups according to IgA anti-β_2_-GP1 positivity: (1) those with positive values (>20.0 CU) and (2) 112 with negative ones (<20.0 CU). 

Verification of definite APS was statistically significantly associated with the positivity for IgA anti-β_2_-GP1 (χ^2^ = 15.00; *p* < 0.0001) (Table 2). The development of APS was 3.84 times higher in patients positive for IgA anti-β_2_-GP1 than in those negative for the latter.

The positive IgA anti-β_2_-GP1 values were associated with thrombosis (χ^2^ = 4.37; *p* = 0.04) (Table 2). The risk of thrombosis was twice as high in patients positive for IgA anti-β_2_-GP1 than in those negative for the latter. There is a significant relationship between IgA anti-β_2_-GP1 and arterial thrombosis. A history of this complication was seen in 26 (41%) patients positive for IgA anti-β_2_-GP1 and in 32 (26%) patients negative for the latter (χ^2^ = 4.67; *p* = 0.03). The risk of arterial thrombosis was 2.04 times higher in patients with IgA anti-β_2_-GP1.

The diagnostic efficiency of IgA anti-β_2_-GP1 was evaluated according to the ROC curves (Figure 4) depending on the presence of thrombosis, pregnancy morbidity, and a confirmed diagnosis of APS. The area under the ROC curve was 0.739 [0.678–0.799] in thrombosis, 0.743 [0.651–0.835] in pregnancy morbidity, and 0.813 [0.759–0.867] in definite APS. All the results according to the ROC curves were significant (*p* = 0.0001).

There was a high specificity of IgA anti-β_2_-GP1 for the diagnosis of APS itself (93%) and its clinical manifestations (90% for thrombosis and 93% for pregnancy morbidity). The sensitivity of IgA anti-β_2_-GP1 for APS, thrombosis, and pregnancy morbidity was 44, 33, and 93%, respectively.

The IgA anti-β_2_-GP1 positivity was combined with anti-β_2_-GP1 and more often with IgG anti-β_2_-GP1 in 100% of cases. The frequency of the IgA anti-β_2_-GP1 positivity was associated more with IgG aCL and IgA anti-β_2_-GP1. A significant correlation was found between IgA anti-β_2_-GP1 and IgG/IgM aCL, IgG/IgM anti-β_2_-GP1. There was no isolated IgA anti-β_2_-GP1 positivity in any patient.

A high correlation was established between IgA anti-β_2_-GP1 and IgA aCL (Figure 5).

## 3. Discussion 

Among the noncriteria aPL, IgA aPL, the diagnostic values, which is currently subject to great debate, are no less interesting for study. 

In our study, the frequency of positive IgA aCL levels in patients with PAPS, probable APS, SLE + APS, and SLE was 49%, 42%, 59%, and 14%, respectively. IgA anti-β_2_-GP1 in patients with these conditions occurred in 41%, 33%, 47%, and 14%, respectively. In addition, IgA aPL was identified in 2% and 1% of cases in the control group (*n* = 100). A slightly higher frequency was found in the general population in the study conducted by Hu et al. [9]. The authors revealed that the prevalence of IgA aCL and IgA anti-β_2_-GP1 in the general population was 2.48% and 2.13%, respectively. According to other studies, the prevalence of IgA aCL ranges from 1.6 to 10% [10,11,12,13], while IgA anti-β_2_-GP1 is from 3 to 20.8% [12,13,14,15]. Among the patients with APS, the incidence of IgA aCL and IgA anti-β_2_-GP1 was higher than that in the study by Hu et al. [9]; their findings showed that the patients with PAPS had IgA aCL in 11.76% of cases and IgA anti-β_2_-GP1 in 10.46%, and these antibodies among the patients with secondary APS were 25.42% and 20.34%, respectively. Authors concluded that the combination of the IgA isotype and the IgG/IgM isotype did not increase the diagnostic performance when compared with the IgG/IgM isotype of aCL or anti-β_2_-GP1, respectively. These values were more than twice lower than ours: IgA aCL and IgA anti-β_2_-GP1 (42% and 41%) in patients with PAPS and these antibodies (59% and 47%) in those with secondary APS. The frequency of IgA aPL in our study was also higher than that in the works by O. Unlu et al. [16], M. Frodlund et al. [17], and M.L. Bertolaccini et al. [18]. On the contrary, the study by A. Vlagea et al. [19] revealed IgA anti-β_2_-GP1 in 76.2% of the patients with SLE. These differences in the data is most likely to be associated with differences in patient populations and with the methods used to determine IgA aPL.

Our data did not add new information regarding the detection of isolated IgA aPL positivity in patients with negative values of classical aPL. However, IgA aPL was detected more frequently in our cohort compared to IgM aPL. An association between IgA aPL and thrombosis was found. The risk of thrombosis was 2.04 times higher with positive IgA aCL compared to patients with negative IgA aCL (OR, 2.04; 95% CI, 1.08–3.84). IgA anti-β_2_-GP1 positive values had twice the risk of thrombosis compared to patients with negative IgA anti-β_2_-GP1 levels (OR, 2.00; 95% CI, 1.04–3.84). The association of IgA aPL with thrombotic events was also confirmed in the work by A. Tsutsumi et al. [20]. When calculating the OR, Y.M. Shen et al. [21] showed that the risk of thrombosis in patients positive for IgA aPL was 1.77 times higher than in those negative for IgA aPL.

When evaluating thrombosis by localization, we found a significant relationship between IgA anti-β_2_-GP1 and arterial thrombosis (χ^2^ = 4.67; *p* = 0.03). The risk of arterial thrombosis in patients with IgA anti-β_2_-GP1 was twice as high than in those without these antibodies. Unlike our results, G. Lakos et al. [22] and S.S. Lee et al. [23] noted an association of IgA anti-β_2_-GP1 positivity with venous thrombosis. On the contrary, the results of the study by R. Ruiz-Garcia et al. [24] agree with those of our study showing a significant preponderance of arterial thrombosis in patients with IgA anti-β_2_-GP1. This is consistent with the data obtained by T. Mehrani et al. [25], who revealed an association between IgA anti-β_2_-GP1 and acute cerebrovascular accident.

In addition to thrombosis, the effect of IgA aPL on the course of pregnancy remains controversial. IgA anti-β_2_-GP1 are assumed to be more associated with pregnancy morbidity than with IgA aCL [26]. This conclusion was reached by R.M. Lee et al. based on a weaker correlation between IgA anti-β_2_-GP1, IgA aCL and morbidity during pregnancy compared with IgG and IgM isotypes. According to A. Tsutsumi et al. [20], the presence of IgA anti-β_2_-GP1 was correlated with presence of LA and/or biological false positive results for serological syphilis test. Levels of IgA anti-β_2_-GP1 significantly correlated with values of IgG anti-β_2_-GP1, IgM anti-β_2_-GP1 but weakly correlated with IgA aCL. The results of our work contradict these data: there is a high correlation between IgA anti-β_2_-GP1 and IgA aCL (R = 0.95; *p* > 0.0001). We did not find a relationship between IgA aPL and pregnancy morbidity either. No association of IgA aPL with pregnancy morbidity in our study may have been due to a small sample size and a retrospective analysis rather than the data obtained during pregnancy. It is also worth noting that we have identified a high specificity of IgA aPL for pregnancy morbidity: 93% for both IgA aCL and anti-β_2_-GP1. The sensitivity of these antibodies was low and amounted to 32% and 29%, respectively. When the ROC curves were constructed for pregnancy morbidity, the AUC of IgA aCL was 0.806 (*p* = 0.0001) and that of IgA anti-β_2_-GP1 was 0.743 (*p* = 0.0003). Thus, further prospective studies are needed to clarify the relationship between IgA aPL and pregnancy morbidity with a larger sample.

Since IgA aPL are predominantly secretory immunoglobulins, R.M. Lee et al. [26] have suggested that the study of IgA aCL and IgA anti-β_2_-GP1 in the cervical and uterine mucosa may be a useful method for the examination of patients with recurrent miscarriages and intrauterine fetal death.

The significance of isolated IgA aPL positivity deserves a special attention. The results of the work by Hu et al. [9] indicate that IgA aPL are usually combined with IgG/IgM aPL, while the isolated positivity is rare in aPL-positive patients (0.29%). Our findings are consistent with the authors’ data on isolated IgA aCL and IgA anti-β_2_-GP1 positivity: no isolated positivity for these antibodies was detected among the examined 187 patients. Several studies [17,19,27] showed isolated IgA aPL positivity that ranged from 1% to 76.2%; however, it was not associated with the clinical manifestations of APS. 

B. Yang et al. [27] noted in their study that IgA aCL and IgA anti-β_2_-GP1 had low diagnostic significance for APS in accordance with the ROC curves: AUC was 0.586 and 0.664, respectively. The same data were obtained by C. Hu et al. [9]: the AUC was 0.670 for IgA aCL and 0.654 for IgA anti-β_2_-GP1. Our findings differed significantly from the results described. Thus, the AUC for IgA aCL was 0.851 (*p* < 0.0001), which for IgA anti-β_2_-GP1 was 0.813 (*p* < 0.0001). The similar results were identified by T. Liu et al. [28]: the AUC for the diagnosis of APS reached 0.814 (for IgA aCL) and 0.778 (for IgA anti-β_2_-GP1).

This work has several limitations. Firstly, that was a monocenter study, including a population of patients who was under medical observation in our center with a reliable diagnosis of APS with and without SLE. We have given a comparison group that was heterogeneous in rheumatic diseases (*n* = 40) and 47.5% of them had episodes of thrombosis. However, this is a real clinical practice. These patients were tested because their doctors wanted to rule out the possible presence of APS. Patients had clinical manifestations of APS (thrombosis and fetal loss), but repeated aPL studies showed negative results. The control group included 100 healthy people. In an ideal study, screening a large population would require attracting more people with single positive results for IgA aPL. This requires resources beyond the capabilities of a single center. Secondly, another limitation is that the LA was not conducted for all patients, since most patients received anticoagulants.

## 4. Subjects and Methods

The study enrolled 187 patients who were followed at the V.A. Nasonova Research Institute of Rheumatology in 2019 to 2021 and who had one of the following diagnoses: PAPS, probable APS, SLE with APS, and SLE without APS (Table 3). The comparison group consisted of patients who were referred to a polyclinic with suspected APS. The comparison group consisted of 49 patients; among them, there were 40 patients with other rheumatic diseases (RDs), 3 pregnant women without RDs with fetal loss in their history, and 6 patients with a history of thrombosis without an established cause. The comparison group was established to assess the incidence of IgA aPL in patients with RDs with and without thrombosis, as well as in patients without RDs who had a history of obstetric pathology or thrombosis. A total of 19 (47.5%) of 40 patients with RDs had a history of thrombosis. The control group included 100 apparently healthy individuals (without RD, who had no cancers or infectious diseases).

The main group, control group, and comparison group were comparable in age. The main and comparison groups were comparable in terms of disease duration. Patients with SLE were younger than patients with primary APS (*p* = 0.01) and with SLE + APS (*p* = 0.001). Patients with primary APS had a shorter duration of illness compared with patients with SLE + APS (*p* = 0.02), but a longer duration compared with patients with probable APS (*p* = 0.04) and with SLE (*p* = 0.03). Disease duration was shorter in patients with probable APS compared with patients with SLE + APS and with SLE (*p* < 0.05) and shorter in patients with SLE compared with patients with SLE + APS (*p* < 0.0001). Women predominated in all groups (Table 3).

The diagnosis of APS was based on the 2006 international classification criteria [3]. PAPS was verified in a patient in the absence of signs of any other disease and in the presence of those of definite APS. The diagnosis of probable APS was made in the absence of signs of rheumatic diseases and a pathology that contributes to the generation of aPL with persistent aPL positivity and/or in the presence of extra-criteria manifestations of the disease (livedo reticularis, thrombocytopenia, cerebral microangiopathy, etc.) [29]. The diagnosis of SLE was based on the 1997 ACR-97 classification criteria [4]. SLE activity was assessed using the Systemic Lupus Erythematosus Disease activity score (SLEDAI) [30].

The study included 122 patients with SLE, of whom 59 (48.3%) were patients with SLE and with APS and 63 (51.7%) with SLE without APS. Frequency of clinical and laboratory manifestations of SLE in patients with/without APS (considering medical history) is presented in Table 4.

The activity of SLE by the SLEDAI index was 4.0 [2.0–10.0] points, and the SLICC damage index was 0.0 [0.0–2.0] points (Table 5). Most patients both with SLE with APS and without APS had low disease activity; a total of 53 (43%) out of 124 patients. Among patients with very high disease activity (*n* = 10), there were significantly more patients with SLE without APS (90%) compared to patients with SLE + APS (*p* = 0.01). SLICC damage index was higher in patients with SLE with APS-1.0 [0.0–3.0]. 

Damage index compared patients with SLE without APS (*p* = 0.007). All the patients included in the study were examined and received basic therapy in inpatient or outpatient settings at the V.A. Nasonova Research Institute of Rheumatology. All the patients underwent standard clinical, laboratory, and instrumental examinations prior to the study inclusion and during the follow-up.

Very high disease activity was more common in patients with SLE without APS compared with patients with SLE and APS (9 (14%) versus 1 (2%), *p* = 0.001). The absence of organ damage according to DI SLICC was associated with SLE without APS: 42 (67%) versus 22 (37%), *p* = 0.001) (Table 5).

The study of aPL involved the determination of IgG/IgM aCL and IgG/IgM anti-β_2_-GP1 by enzyme-linked immunoassay (ELISA), IgG/IgM/IgA aCL and IgG/IgM/IgA anti-β_2_-GP1 by chemiluminescent assay (CLA), LA. 

The patients included in the study, the patients of the comparison group, and the control group were tested for IgG/IgM/IgA aCL, IgG/IgM/IgA anti-β_2_-GP1 by CLA using a BIO-FLASH^®^ analyzer (Biokit S.A., Spain). The reagent kits were AcuStar (Spain) for the detection of IgG/IgM anti-β_2_-GP1 and IgG/IgM aCL and QUANTA Flash^®^ (USA) for determination of IgA aCL, IgA anti-β_2_-GP1 and IgG anti-β_2_-GP1DI. The tested aPLs were measured in chemiluminescent units (CU). Based on the mean values of the control group for the determination of IgG/IgM/IgA aCL, IgG/IgM/IgA anti-β_2_-GP1, the positivity levels were identified according to the formulas: arithmetic mean (M) + 3 or 5 standard deviations (SDs): M + 3 SDs and M + 5 SDs. The diagnostic significance of the isolated levels for positivity and the levels proposed by the reagent manufacturers was assessed, as a result of which the positivity levels were determined: for IgG aCL > 25.9 CU (M + 5 SDs), for IgM aCL > 19.5 CU (M + 3 SDs), for IgA aCL > 18.9 CU (M + 5 SDs), for IgG anti-β_2_-GP1 > 32.0 CU (M + 5 SDs), for IgM anti-β_2_-GP1 > 6.9 CU (M + 3 SDs), and for IgA anti-β_2_-GP1 > 20.0 CU (the data from the reagent’s manufacturer). 

Statistical analysis. The following indicators were used to describe quantitative variables: arithmetic mean (M), standard deviation (δ), median, 25th and 75th percentiles, as well as frequency for qualitative variables. Differences were considered to be statistically significant at *p* ≤ 0.05. The frequency differences for two independent study group objects were analyzed using statistical tests: Pearson’s chi-squared (χ^2^) test; a χ^2^ value of less than 10 was employed with the Yates correction if there were absolute frequencies in the cells of frequency tables. To analyze the diagnostic efficiency of laboratory tests, the ROC “error curve” was applied, which reflected the relationship between the frequency of true positive results and that of false positive results. ROC curves were created using the IBM SPSS Statistics 13.0 for Windows software package (IBM Corporation, Armonk, NY, USA). The clinical informative value of a laboratory test was determined by how high its ROC curve was. To evaluate the ROC curves, the area under the curve (AUC = area under the curve) was calculated. AUC was estimated in the range of 0–1: <0.6 (unsuitable), 0.61–0.8 (revision required), ≥0.81 (admissible for clinical validation) [31,32].

Calculation was made on a personal computer using the statistical data analysis package Statistica 10.0 for Windows (StatSoft Inc., USA) and IBM SPSS Statistics 13.0 for Windows (IBM Corporation, Tulsa, OK, USA), VassarStats.

## 5. Conclusions

There was no isolated positivity of IgA antiphospholipid antibodies in our group of patients. At the same time, the presence of IgA aCL and IgA anti-β_2_-GP1 was associated with thrombosis, while significantly associated with arterial thrombosis and significant APS. Despite some limitations in this work concerning the heterogeneity of the comparison group, the obtained data indicate the need for further studies of IgA aPL in different patient populations.

## Figures and Tables

**Figure 1 ijms-23-09432-f001:**
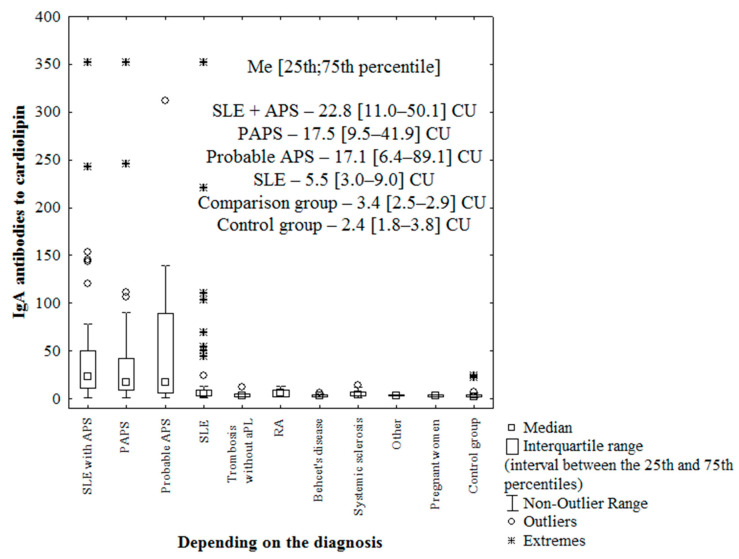
Levels of IgA antibodies to cardiolipin (IgA aCL) in patient groups. Note: SLE-systemic lupus erythematosus; APS-antiphospholipid syndrome; aPL-antiphospholipid antibodies; RA-rheumatoid arthritis; other-two patients with polymyositis, one of whom had thrombosis and one patient with Buerger’s endarteritis; Me-median with interquartile range; CU-chemiluminescent units; and the Y axis shows units-CU-chemiluminescent units (IgA aCL unit).

**Figure 2 ijms-23-09432-f002:**
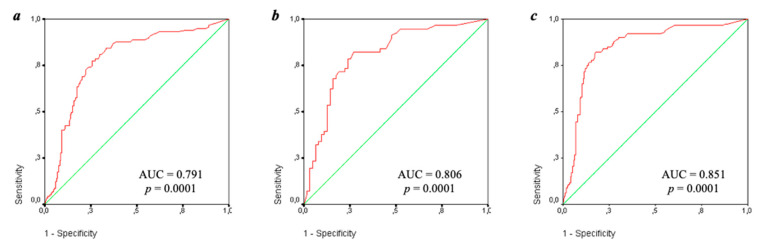
ROC-curves IgA aCL depending on thrombosis (**a**), obstetric pathology (**b**), and reliable antiphospholipid syndrome (**c**). Note: AUC-area under curve, *p*–reliability.

**Figure 3 ijms-23-09432-f003:**
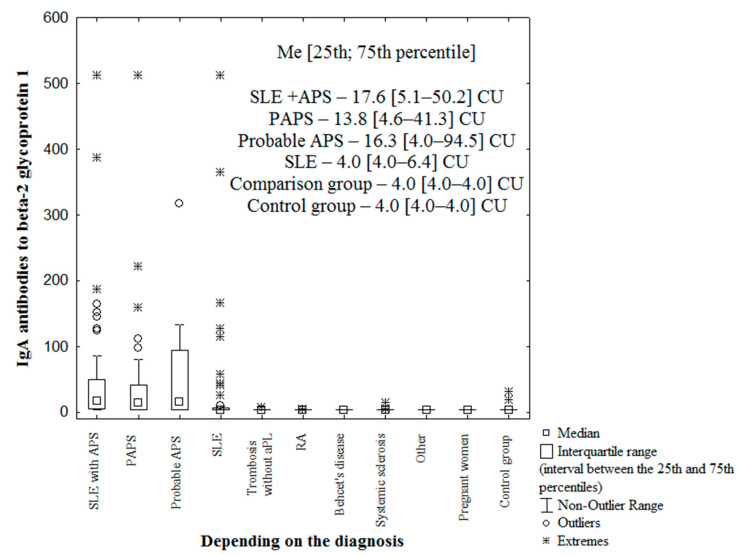
Levels of IgA antibodies to beta-2 glycoprotein 1 (IgA anti-β_2_-GP1) in patient groups. Note: SLE-systemic lupus erythematosus; APS–antiphospholipid syndrome; aPL-antiphospholipid antibodies; RA-rheumatoid arthritis; other-two patients with polymyositis, one of whom had thrombosis and one patient with Buerger’s endarteritis; Me-median with interquartile range; CU-chemiluminescent units; and the Y axis shows units-CU-chemiluminescent units (IgA anti-β_2_-GP1 unit).

**Figure 4 ijms-23-09432-f004:**
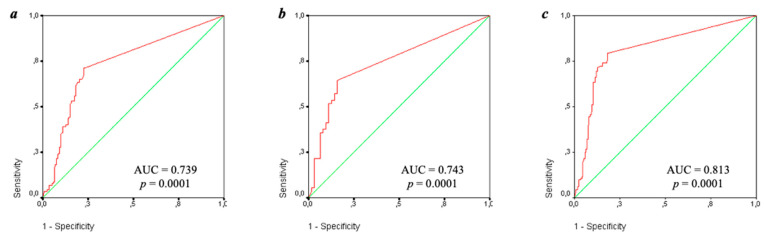
ROC-curves IgA anti-β_2_-GP1 depending on thrombosis (**a**), obstetric pathology (**b**), and reliable antiphospholipid syndrome (**c**). Note: AUC-area under curve and *p*-reliability.

**Figure 5 ijms-23-09432-f005:**
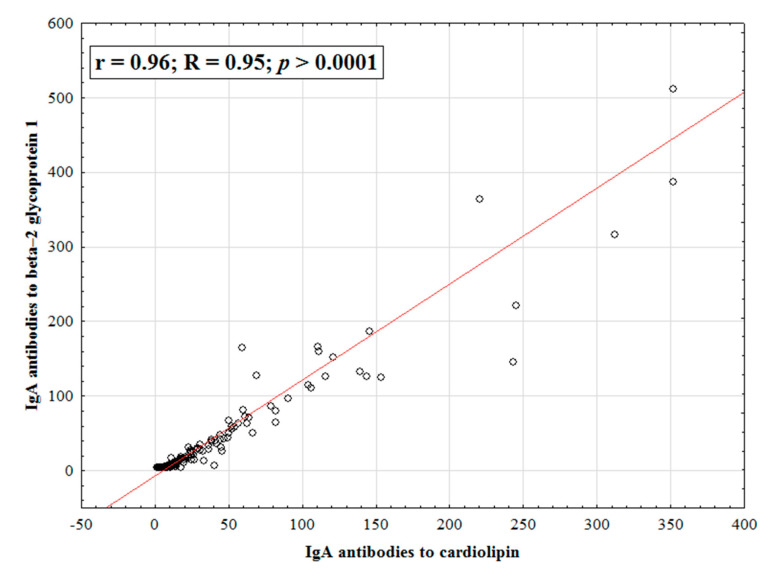
Correlation of IgA antibodies to beta-2 glycoprotein 1 with IgA antibodies to cardiolipin. Note: r-Pearson correlation coefficient, R-Spearman correlation coefficient, and *p*-reliability.

**Table 1 ijms-23-09432-t001:** Relationship of antiphospholipid syndrome and it is clinical manifestations (thrombosis and pregnancy morbidity) with IgA antibodies to cardiolipin (IgA aCL).

Parameter		IgA aCL (+), *n* (%)	IgA aCL (−), *n* (%)	χ^2^; *p* OR [95% CI]
**APS**	yes	61 (81)	51 (45)	**23.96; <0.0001** **5.26** **[** **2.63–11.11** **]**
	no	14 (19)	61 (55)	
**Thrombosis** **(total)**	yes	53 (71)	61 (54)	**4.96; 0.02** **2.04 [1.08–3.84]**
	no	22 (29)	51 (46)	
**Arterial**	yes	29 (39)	29 (26)	3.43; 0.06 1.81 [0.97–3.44]
	no	46 (61)	83 (74)	
**Venous**	yes	36 (48)	45 (40)	1.12; 0.28 1.38 [0.76–2.50]
	no	39 (52)	67 (60)	
**Pregnancy morbidity ***	yes	*n* = 25 20 (80)	*n* = 45 34 (75)	0.02; 0.88 1.29 [0.39–4.34]
	no	5 (20)	11 (25)	

Note: total–total number of thrombosis regardless of localization; * pregnancy morbidity, calculated from the number of women who had pregnancy against the background of the disease; *n*-number of patients; χ^2^-agreement criterion; *p*-reliability; OR-odds ratio; and CI-confidence interval.

**Table 2 ijms-23-09432-t002:** Relationship of antiphospholipid syndrome and it is clinical manifestations (thrombosis and pregnancy morbidity) with IgA antibodies to beta-2 glycoprotein 1 (IgA anti-β_2_-GP1).

Parameter		IgA anti-β_2_-GP1 Positive, *n* (%)	IgA anti-β_2_-GP1 Negative, *n* (%)	χ^2^; *p* OR [95% CI]
APS	yes	50 (79)	62 (50)	15.00; 0.0001 3.84 [1.92–8.33]
	no	13 (21)	62 (50)	
Thrombosis (total)	yes	45 (71)	69 (56)	4.37; 0.04 2.00 [1.04–3.84]
	no	18 (29)	55 (44)	
Arterial	yes	26 (41)	32 (26)	4.67; 0.03 2.04 [1.06–3.84]
	no	37 (59)	92 (74)	
Venous	yes	31 (49)	50 (40)	1.34; 0.24 1.44 [0.78–2.70]
	no	32 (51)	74 (60)	
Pregnancy morbidity *	yes	*n* = 21; 18 (86)	*n* = 49; 36 (73)	0.02; 0.88 1.31 [0.40–4.34]
	no	5 (14)	13 (27)	

Note: total–the total number of thrombosis regardless of localization; * pregnancy morbidity, calculated from the number of women who had pregnancy in their disease course; *n*-number of patients; χ^2^-agreement criterion; *p*-reliability; OR-odds ratio; and CI-confidence interval.

**Table 3 ijms-23-09432-t003:** The clinical and laboratory characteristics of patients included in the study.

Parameter	PAPS, *n* = 53	Probable APS, *n* = 12	SLE + APS, *n* = 59	SLE, *n* = 63	Total, *n* = 187	Comparison Group, *n* = 49	Control Group, *n* = 100
Average age, Me [25;75 percentile], years	38.0 [32.0–43.0]	34.0 [29.5–44.5]	40.0 [33.0–47.0]	31.0 [24.0–41.0]	39.0 [35.0–48.0]	39.0 [35.0–48.0]	41.0 [30.0–54.0]
Duration of the disease, Me [25;75 percentile], years	8.4 [3.1–13.5]	0.9 [0.3–2.1]	15.0 [7.3–21.0]	4.0 [1.5–8.0]	7.0 [2.0–15.0]	6.0 [1.6–13.0]	-
Sex: male/female, abs	30/23	10/2	47/12	55/8	142/45	33/16	86/14
History of thrombosis, abs (%)	48 (91)	1 (8) ***	51 (86)	14 (22)	19 (39)	19 (39)	1 (1)
Obstetric pathology *, *n* (%)/*n*	20 (95)/21	1 (50) ****/2	26 (81)/32	7 (44)/16	5 (15)/33	5 (15)/33	2 (4); *n* = 51
Thrombocytopenia for the entire period of the disease, *n* (%)	7 (13)	5 (42)	19 (32)	15 (24)	0 (0)	0 (0)	0 (0)
Livedo reticularis, *n* (%)	10 (19)	2 (17)	19 (32)	5 (8)	0 (0)	0 (0)	0 (0)
IgG aCL, *n* (%)	42 (79)	6 (50)	52 (88)	13 (21)	0 (0)	0 (0)	0 (0)
IgM aCL, *n* (%)	19 (36)	5 (42)	18 (30)	10 (16)	2 (4)	2 (4)	4 (4)
IgG anti-β_2_-GP1, *n* (%)	41 (77)	6 (50)	52 (88)	18 (29)	6 (12)	6 (12)	1 (1)
IgM anti-β_2_-GP1, *n* (%)	19 (36)	8 (67)	18 (30)	12 (19)	3 (6)	3 (6)	2 (2)
Lupus anticoagulant **, *n* (%)/*n*	7 (87.5)/8	8 (89)/9	10 (71)/14	19 (79)/24	0 (0); *n* = 1	0 (0); *n* = 1	-
Therapy:							
Glucocorticoids	4 (7)	2 (17)	47 (80)	53 (84)	106 (57)	31 (63)	-
Hydroxychloroquine	31 (58)	4 (33)	52 (88)	54 (86)	141 (75)	8 (16)	
Basic anti-inflammatory drugs	0 (0)	0 (0)	7 (12)	18 (29)	25 (13)	26 (53)	
Aspirin	17 (32)	3 (25)	24 (41)	18 (29)	62 (33)	8 (16)	
Low-molecular-weight heparin	7 (13)	1 (8)	10 (17)	14 (22)	32 (17)	8 (16)	
Direct Oral anticoagulants	27 (51)	1 (8)	18 (31)	2 (3)	48 (26)	8 (16)	
Warfarin	11 (21)	0 (0)	14 (24)	2 (3)	27 (15)	1 (2)	
Without anticoagulant therapy	8 (15)	10 (84)	16 (27)	45 (72)	79 (42)	32 (66)	

Note: IgG/IgM aCL and IgG/IgM anti-β_2_-GP1 were determined by chemiluminescence assay; * obstetric pathology was calculated in women who had pregnancy in their disease course, in the numerator–number and % of women with obstetric pathology, in the denominator–number of women who had pregnancy in their disease course; ** lupus anticoagulant study was performed in patients who did not receive anticoagulant therapy; numerator is the number and % of patients with positive lupus anticoagulant, and denominator is the number of patients who had lupus anticoagulant determination; *** patients with a history of seven thromboses (if other causes are excluded) and livedo reticularis; **** patients with one pregnancy resulting in fetal loss before 10 weeks of gestation, thrombocytopenia, and persistent high antiphospholipid antibody positivity; in the control group, one had post-injection thrombosis, one had fetal loss before the 10th week of gestation, and one had preeclampsia; *n*—number of patients, Me [25;75 percentile]–median with interquartile range, abs—absolute values, SLE—systemic lupus erythematosus, APS—antiphospholipid syndrome, aCL—antibodies to cardiolipin, anti-β_2_-GP1–antibodies to beta-2 glycoprotein 1.

**Table 4 ijms-23-09432-t004:** Clinical and laboratory manifestations of systemic lupus erythematosus over the entire period of the disease according to 1997 criteria.

Parameters	SLE with APS (*n* = 59) *n* (%)	SLE without APS (*n* = 63) *n* (%)	Total (*n* = 122) *n* (%)
Erythema of the zygomatic arches	19 (32)	26 (41)	45 (37)
Discoid rash	3 (5)	2 (3)	5 (4)
Photosensitization	11 (19)	14 (22)	25 (20)
Oral ulcers	8 (13)	18 (28)	26 (21)
Arthritis	29 (49)	39 (62)	68 (56)
Serositis	27 (46)	32 (51)	59 (48)
Renal damage	22 (37)	27 (43)	49 (40)
Central nervous system damage	9 (15) *	1 (2)	10 (8)
Hematological disorders	38 (64)	44 (70)	82 (67)
Immunological disorders	57 (97)	62 (98)	119 (97)
Positive antinuclear factor	59 (100)	63 (100)	122 (100)
IgA anti-β_2_-GP1 positivity	28 (47) **	9 (14)	37 (30)
IgA aCL positivity	35 (59) **	9 (14)	44 (36)

Note: * Central nervous system lesions were significantly more common in patients with SLE + APS compared to patients with SLE without APS (*n* = 0.01); ** IgA anti-β_2_-GP1 and IgA aCL (at the time of inclusion in the study) were significantly more common in patients with SLE + APS compared to patients with SLE without APS (*p* = 0.0002 and *p* < 0.0001, respectively).

**Table 5 ijms-23-09432-t005:** Assessment of activity and organ damage in patients with systemic lupus erythematosus at the time of study inclusion.

SLEDAI			
	SLE with APS (*n* = 59) *n* (%)	SLE without APS (*n* = 63) *n* (%)	Total (*n* = 122) *n* (%)
No activity	7 (12)	7 (11)	14 (11)
Low activity	30 (49)	23 (36)	53 (43)
Medium activity	14 (25)	12 (19)	26(22)
High activity	7 (12)	12 (19)	19 (16)
Very high activity	1 (2)	9 (14)	10 (8)
SLEDAI activity index (points; median, 25th and 75th percentiles)	4.0 [2.0–8.0]	6.0 [2.0–14.0] *	4.0 [2.0–10.0]
SLICC Damage index			
No damage	22 (37)	42 (67)	64 (57)
Low DI	16 (27)	8 (13)	24 (17)
Medium DI	19 (32)	11 (17)	30 (25)
High DI	2 (3)	2 (3)	4 (3)
SLICC Damage index (Points; median, 25th and 75th percentiles)	1.0 [0.0–3.0] **	0.0 [0.0–1.0]	0.0 [0.0–2.0]

Note: * Patients with SLE without APS had a higher index of disease activity compared to patients with SLE with APS (*p* = 0.006); ** patients with SLE with APS had higher SLICC.

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
