# Peer review of "IgA Antiphospholipid Antibodies in Antiphospholipid Syndrome and Systemic Lupus Erythematosus"

_ijms, 2022, doi:10.3390/ijms23169432_

Round 1

Reviewer 1 Report

Interesting research on the significance of IgA antiphospholipid antibodies in patients with different cases of APS.

These are my remarks:

1 1.  For a definete diagnosis of APS, the persistent presence of moderate to high serum levels of antiphospholipid antibodies (aPL) is necessary. The sentence is unnecessary - throw it out.

2 2.    According to the latest international classification criteria, the serological markers of APS include IgG and/or IgM anti-cardiolipin antibodies (aCL) in serum or plasma, which are present in medium or high levels (>40 GPL or MPL units), IgG and/or IgM anti-beta-2 glycoprotein 1 antibodies (anti- β2-GP1), and lupus anticoagulant (LA), which are detectedtwo or more times at a study time interval of at least 12 weeks.- A very complicated sentence-rephrase

3 3. This leads to a significant delay in the start of certain treatment, especially in emergency situations.- delete the sentence because it is not necessary to wait for confirmation of autoantibodies to apply therapy in emergency situations

4  4. Table 1. The clinical and laboratory characteristics of 187 patients included in the study : the content of the table does not correspond to the title, the comparison groups and the control group are listed in the table. Did the mentioned groups also have positive IgG and IgM antiphospholipid antibodies - as can be concluded from the table? The table should be completely rearranged and relevant data should be shown: are the patients and control groups matched in terms of age and sex, indicate what therapy the subjects were taking (immunosuppressive therapy can affect the level of antibodies)

5   5. The comparator group is very heterogeneous and not necessary

6   6. Table 2. Clinical and laboratory manifestations of systemic lupus erythematosus over the entire period of the disease according to 1997 criteria - instead of unnecessarily stating the clinical manifestations of SLE with and without APS, I think it would be interesting to see if there are significant differences in the manifestations of these two groups of patients or to completely remove the table.

7  7. Table 3. Assessment of activity and organ damage in patients with systemic lupus erythematosus at the time of study inclusion-  it should be stated whether there is any significance in the differences in activity and impairment index between the mentioned groups of patients

8   8.  APS was detected in 61 (81%) patients with IgA aCL versus 51 patients without these antibodies (χ2 = 23.96; p < 0.0001; odds ratio (OR), 5.26; 95% confidence interval (CI), 2.63-11.11) -the percentage behind 51 is missing

9   9. Although in our cohort of patients there was a more frequent detection of IgA aPL and the relationship between IgA aPL and thrombosis was found, the risk of thrombosis with positive values of IgA aCL was 2.04 times higher, and with positive values of IgA anti-β2-GP1 it was twice as high- Rephrase the sentence - write comprehensibly

1  10. Conclusion.-without point; reformulate and simplify the conclusion, without repeating data from the results

1  11. Check the punctuation

Author Response

Author's Reply to the Review Report (Reviewer 1)

Reviewer 2 Report

-abstract :

*only p <0.05, result is missing

-definition of probable APS : extra criteria or only biologic antiphospholipid?

-criteria for comparison group? 187 vs 49?

-treatment : aspirine, anticoagulant

-results in subject and method part

-hard to follow because of new groups in the results, not described in methods. Table 4 need more details about the group. 

-same for table 5

Author Response

Dear colleague, thank you so much for your work on our article. I agree with your comments and have made corrections in the text. I also attached our responses to the comments

Reviewer 3 Report

This study demonstrated a relationship between thrombosis and APS with IgA aCL and IgA anti-β2-GP1. It is well designed and data are presented logically.

It is worthwhile to be published.

Author Response

Dear colleague, thank you so much for your work on our article. 

Round 2

Reviewer 1 Report

1.     Tables are not corrected as recommended. However, since the significance levels for key data are set out in the rationale, I accept the changes. However, I still think the conclusion needs to be improved. The limitations of the study should be highlighted (incoherent comparator group.....)

 limitations of the study should be highlighted (incoherent comparator group.....)

Author Response

Dear Reviewer, Thank you so much for reviewing our article again. In the discussion on page 18, I added some limitations of our work. This, of course, concerns the comparison group. In the article, I highlighted it in green.

This work has several limitations. Firstly, this is a monocenter study, including a population of patients under observation in our center with a reliable diagnosis of APS with and without SLE. We have given a comparison group that was heterogeneous in rheumatic diseases (n=40) and 47.5% of them had episodes of thrombosis. However, this is a real clinical practice. These patients were tested because their doctors wanted to rule out the possible presence of APS. Patients had clinical manifestations of APS (thrombosis and fetal loss), but repeated aPL studies showed negative results. The control group included 100 healthy people. In an ideal study, screening a large population would require attracting more people with single positive results for IgA-aPL. This requires resources beyond the capabilities of a single center. Secondly, another limitation is that the LA was not conducted for all patients, since most patients took anticoagulants.

I am very sorry, table 1 on the characteristics of patients has been completely redone and statistically significant differences in the groups have been noted.
